# A systematic review of statistical methodology used to evaluate progression of chronic kidney disease using electronic healthcare records

Faye Cleary⦿*, David Prieto-Merino‡, Dorothea Nitsch‡

Department of Non-Communicable Disease Epidemiology, London School of Hygiene and Tropical Medicine, London, United Kingdom

‡ DPM and DN also contributed equally to this work.
* faye.cleary@lshtm.ac.uk

**Data Availability Statement:** This is a systematic review of previously published research, available in the public domain. All relevant data extracted

## Abstract

### Background

Electronic healthcare records (EHRs) are a useful resource to study chronic kidney disease (CKD) progression prior to starting dialysis, but pose methodological challenges as kidney function tests are not done on everybody, nor are tests evenly spaced. We sought to review previous research of CKD progression using renal function tests in EHRs, investigating methodology used and investigators' recognition of data quality issues.

### Methods and findings

We searched for studies investigating CKD progression using EHRs in 4 databases (Medline, Embase, Global Health and Web of Science) available as of August 2021. Of 80 articles eligible for review, 59 (74%) were published in the last 5.5 years, mostly using EHRs from the UK, USA and East Asian countries. 33 articles (41%) studied rates of change in eGFR, 23 (29%) studied changes in eGFR from baseline and 15 (19%) studied progression to binary eGFR thresholds. Sample completeness data was available in 44 studies (55%) with analysis populations including less than 75% of the target population in 26 studies (33%). Losses to follow-up went unreported in 62 studies (78%) and 11 studies (14%) defined their cohort based on complete data during follow up. Methods capable of handling data quality issues and other methodological challenges were used in a minority of studies.

### Conclusions

Studies based on renal function tests in EHRs may have overstated reliability of findings in the presence of informative missingness. Future renal research requires more explicit statements of data completeness and consideration of i) selection bias and representativeness of sample to the intended target population, ii) ascertainment bias where follow-up depends on risk, and iii) the impact of competing mortality. We recommend that renal progression

from reviewed articles are captured in the manuscript and its supporting Information files.

**Funding:** This work was supported by the Medical Research Council (MR/N013638/1), grant awarded to FC. The funders had no role in study design, data collection and analysis, decision to publish, or preparation of the manuscript.

**Competing interests:** The authors have declared that no competing interests exist.

studies should use statistical methods that take into account variability in renal function, informative censoring and population heterogeneity as appropriate to the study question.

## Introduction

Chronic kidney disease (CKD) is a growing public health problem [1, 2]. Risks associated with CKD include cardiovascular morbidity, death, and in rare cases progression to end-stage renal disease (ESRD) requiring renal replacement therapy (RRT) [3]. Severity of disease, mechanism of renal damage and rate of progression of disease vary between patients, and the disease may change course over time in response to changing risk factors [4, 5]. While a minority of patients progress to ESRD, the cost of RRT presents a substantial economic burden to public health services and is likely to increase further over the coming years as prevalence of RRT rises alongside population growth and an ageing population [6, 7]. Increasing adoption of electronic healthcare records (EHRs) offers an opportunity to study progression of kidney disease in real-world care, that may enable improved decision-making in clinical practice. Whilst there is the promise of big sample sizes to be analysed, constraints on data availability of renal function test results may complicate reliable evaluation in EHRs. Frequency of monitoring of renal function is likely to vary in routine care according to differing individual patient risk profiles, local healthcare policy, physician-related factors, area of management within the healthcare system, social factors, or temporary illness. This may lead to some members of the target population being less likely to be followed up for renal function, potentially leading to selection and ascertainment biases in the study of CKD progression that may result in unreliable conclusions.

There are other methodological challenges in evaluation of CKD progression that are not specific to EHRs that should be considered by researchers. Deterioration in renal function over time is most commonly detected through changes of the estimated glomerular filtration rate (eGFR), usually derived from serum creatinine, sex, age, and ethnicity. Such creatinine-based GFR-estimating equations are imprecise, particularly at high levels of eGFR [8, 9]. Major changes in renal function in the context of acute illness are a sign of acute kidney injury (AKI). Although AKI is at least partially reversible in surviving patients, a history of AKI may accelerate subsequent loss in renal function. However, when researchers study eGFR decline over time, often statistical models are used that ignore the impact of acute drops in renal function on the subsequent trajectory. Population heterogeneity (caused by variation in risk factors both at baseline and evolving over time) may complicate analyses that assume a common mean linear trajectory of renal function loss over time, and it may be necessary to use more sophisticated methods if this assumption is violated that take this variability into account. Unmeasured confounding may also present issues, particularly if important confounders are not considered in the analysis. Competing events such as initiation of RRT or death complicate evaluation of progression outcomes. A previous systematic review by Boucquemont et al. in 2014 [10] reviewed statistical methods used to identify risk factors for progression of CKD, covering research on cohort studies published between 2002 and 2012. They summarised most used outcome measures and statistical models, critiquing handling of bias due to informative censoring, competing risks, correlation due to repeated measures, and non-normality of response, and proposed recommendations for best practice statistical methods and software packages.

We performed a systematic review of all longitudinal analyses of renal function tests investigating the nature, burden or consequences of CKD progression using EHRs. We aimed to establish how data issues inherent to EHRs and methodological challenges were handled, how CKD progression was defined, what statistical methods were used and whether data issues were acknowledged in the context of reliability of study conclusions.

## Materials and methods

### Protocol and registration

There is no published protocol available for this systematic review. Prior to completion of data extraction, this review was registered in the PROSPERO international prospective register of systematic reviews (registration number CRD42020182587).

### Eligibility criteria

This is a review of statistical methodology covering all research studying the nature, burden or consequences of CKD progression using EHRs. Our intention was to focus on how researchers used renal function tests to study CKD progression. Initiation of dialysis is already a well-established clinically important outcome and as this was not the subject of the review, we excluded dialysis endpoints (as a measure of CKD progression) from review. Populations that had already initiated RRT at baseline or that were sampled on the basis of RRT initiation were excluded from review, since such populations are not appropriate for studying progression of CKD. (This criterion does not exclude patients that initiated RRT during follow-up.) Measures of CKD progression may constitute either exposures or outcomes of analysis. PICOS criteria are listed in the table below. There are no restrictions on sample size, population location or date of publication. Only studies reported in English language are included.

| | |
|---|---|
| Participants | Include: Adults aged ≥18 with CKD stages 3–5; Studies that involve both CKD and non-CKD patients are also included, e.g. diabetes<br>Exclude: Patients who have initiated RRT (dialysis or transplant), even if data is collected for renal function prior to RRT initiation; Patients with AKI (unless chronic changes are also studied); Non-human subjects; Children |
| Intervention/ Exposure | No restriction if CKD progression is measured as the outcome, rather than exposure.<br>If CKD progression is analysed as an exposure, restrictions of this measure apply (see outcome definition). |
| Comparators/ Control | No restriction. |
| Outcome | No restriction on outcomes if CKD progression is measured as an exposure, rather than outcome.<br>If the outcome is a measure of CKD progression:<br>Include: Measures of chronic change in renal function based on multiple measures of eGFR or any other measure that may be used to infer eGFR (e.g. serum creatinine, cystatin-C, iohexol clearance), e.g. rate of change, change from baseline, regression slope, time to change or threshold eGFR<br>Exclude: All other measures of renal function, e.g. proteinuria; Studies of acute AKI or short term follow up (<6 months) of renal function following a procedure; Single time-point analyses; Time to RRT as single outcome. |
| Study design | Include: Retrospective analysis of routinely collected electronic healthcare records which may include retrospective cohort studies, case-control studies and cross-sectional studies (if a measure of past progression is included)<br>Exclude: Case reports, Clinical trials, prospective cohort studies or any other study design with pre-planned data collection strategy for research purposes. |

## Searches

We performed electronic searches of MEDLINE, EMBASE, Global Health and Web of Science databases through to 11[th] August 2021. A copy of the search strategy is provided in the supplementary materials S2 File.

## Study selection

This study had one lead reviewer and two supporting reviewers. The lead reviewer was responsible for screening all articles for eligibility, which involved scrutiny of abstracts followed by full-text review. The two supporting reviewers independently screened a sample of 50 articles each for eligibility. Consistency of agreement and reasons for disagreement were discussed. Clarity of inclusion/exclusion criteria was updated following discussion and prior to completion of eligibility review by the lead reviewer.

## Data collection process

The lead reviewer was responsible for data extraction for all eligible research articles. In addition, key items that were the subject of this review were validated by supporting reviewers who independently extracted the following items for all articles: (1) measure of change in renal function; (2) statistical methods used in analysis of changes in renal function; and (3) definitions of progression of CKD, if any. The lead reviewer developed a data extraction form in an Excel spreadsheet, which was reviewed and approved by supporting reviewers in the initial stages of data extraction.

## Data items

Information extracted from eligible research articles included details of the study population, study methodology and how data quality issues and other methodological issues were handled. Extracted items are listed below.

**Study population.** Data collection timeframe; Country of residence; Mean age; Percent male; Primary morbidity under study / reason for inclusion; Data source / healthcare setting

**Study methodology.** Date of publication; Study design; Research aims; Sample size (before and after exclusions for reasons of data completeness [for details, see below explanation of data completeness inclusion criteria and calculations of percentage of target population analysed]); Measure of renal function; Measure of change in renal function over time; Definition of progression (if any); Whether change in renal function was exposure or outcome; Duration of follow up for changes in renal function; Data completeness inclusion criteria and the minimum number of renal function tests required for analysis; Statistical tools used; Statistical model used.

Some additional results were derived to quantify data completeness for analysis, including the percentage of the target population that were analysed after application of data completeness inclusion criteria and the percentage of patients that dropped out of analysis during the intended follow up period having met criteria for inclusion in analysis. Here, "data completeness inclusion criteria" refer to the study-specific inclusion criteria applied prior to main analyses being performed that aimed to retain only those patients with sufficient data completeness to be deemed suitable for analysis, with such criteria expected to vary between studies.

*Percentage of target population analysed* was defined as:

$$\frac{number\ of\ patients\ analysed\ (meeting\ population\ criteria\ after\ exclusions\ due\ to\ data\ completeness)}{number\ of\ patients\ meeting\ population\ criteria\ prior\ to\ exclusions\ due\ to\ data\ completeness} \times 100$$

This was computable in some but not all studies, as it requires data on the total number of patients included in analysis as well as the number of patients that met population criteria before data completeness exclusion criteria were applied. (In propensity score matched cohort studies, propensity score matching criteria are included in population criteria, and we only compute percentage of target population analysed in the propensity score matched cohort, where this is possible.)

*Percentage of study population lost to follow up* was defined as:

$$\frac{number\ of\ analysed\ patients\ lost\ to\ follow\ up\ during\ the\ intended\ follow\ up\ period}{number\ of\ patients\ analysed} \times 100$$

Again, this was computable in some but not all studies, as it requires data on the number of patients analysed and the number of those patients that dropped out during the intended follow up period, for example due to death, initiation of RRT or other lack of follow up in routine care which could be for many different reasons.

**Handling of data quality issues and other methodological challenges.** Of the items below, details extracted included whether items were mentioned, whether information was provided on data completeness [if relevant], whether implications were acknowledged, whether challenges were tackled methodologically and any statistical methods used to attempt to overcome challenges:

Handling of sample completeness / representativeness of the target population; Handling of informative drop-outs/censoring; Handling of missing longitudinal data; Handling of missing covariate data; Distributional checks/issues; Handling of within-patient correlation and variability of kidney function over time; Handling of population heterogeneity; Handling of confounding.

## Risk of bias in individual studies

Assessment of bias in individual studies was one of the main aims of this systematic review. Key measures of bias evaluated in individual studies were the percentage of the sample target population that were analysed and the percentage of the analysed study population that were lost to follow up. Study-specific measures were reported and bar charts were produced for these measures to demonstrate the potential for bias in individual studies due to informatively missing data.

## Synthesis of results

This review was descriptive with simple aggregation of collected data items only and no statistical analysis was performed. 4 separate summaries are provided to describe study population characteristics, study methodology used, acknowledgment and handling of data quality issues and other methodological challenges, and definitions of CKD progression. For studies exploring multiple outcomes or conducting multiple analyses of changes in renal function, the outcomes and analyses considered the primary focus regarding renal progression in each paper are summarised in the review.

## Risk of bias across studies

There was no single effect size of interest in this study and no meta-analysis was performed, as the review focussed on methodology used and investigators' handling of data quality issues. Publication bias was therefore challenging to evaluate, as funnel plots and statistical tests could not be used. Efforts were made to maximise coverage of peer-reviewed literature in this field, including extraction of articles from 4 major databases. If research is missing from review due to publication

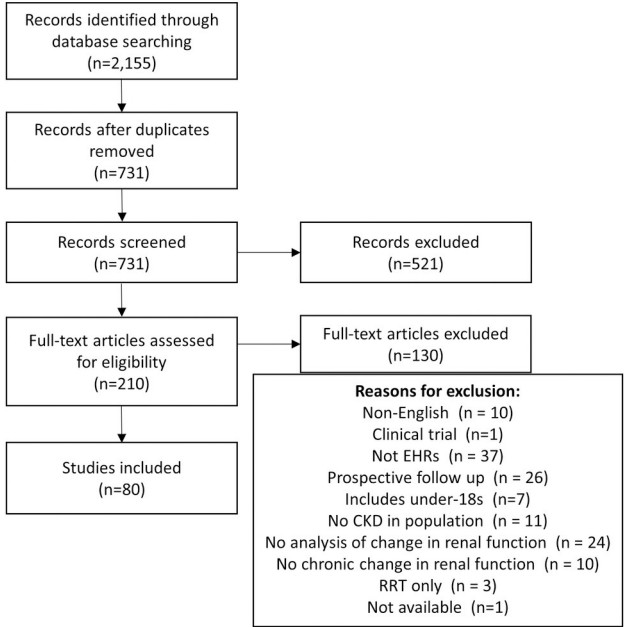

**Fig 1. Flow chart of study selection.**

in non-English languages, then data quality issues in such missing studies are likely to be similar to those in English language studies that were included. There will be clinical audit studies that are not peer-reviewed; these studies are likely to be of a similar of worse quality than reviewed studies because peer-reviewed literature is expected to go through certain research quality checks. In any case, as peer-reviewed literature is more likely to be used to inform policy than other research, this is arguably the optimal collection of research to assess the aims of this review.

## Results

731 unique articles were identified from database searching, of which 80 met study eligibility criteria (Fig 1). Primary reasons for exclusion were not using EHRs, pre-planned data collection for research purposes such as a prospective cohort study, and studies with a single renal function test rather than longitudinal analysis of repeated measures of renal function. Other reasons for exclusion were ineligible populations, such as studies including children, restricted to RRT populations or studies that did not include CKD patients, such as studies of the incidence of CKD. All included studies retrospectively analysed routinely collected healthcare data. It was not always clear whether electronic or paper records were used, and while efforts were taken to differentiate this, it is possible that some included studies may have involved manual data extraction from paper records. 70 studies (88%) clearly stated the use of EHRs. In the 10 studies that did not state this, the time-frame for data collection and location of research suggested that electronic healthcare systems were likely to have been used, but we could not verify this. These studies have been summarised separately in the supplementary materials. A full list of reviewed studies is also included in the supplementary materials S3 File.

### Study population characteristics

Table 1 summarises characteristics of study populations analysed in reviewed articles. Research was most commonly conducted in the UK (25%) and USA (30%), followed by East Asian

**Table 1. Summary of study populations studied (N = 80).**

| Study population characteristics | N (%) |
|---|---|
| Primary decade of follow up | |
| 2010–2019 | 35 (43.8%) |
| 2000–2009 | 36 (45.0%) |
| 1990–1999 | 3 (3.8%) |
| Not available | 6 (7.5%) |
| Country | |
| **Europe** | **28 (35.0%)** |
| UK | 20 (25.0%) |
| Germany | 2 (2.5%) |
| Italy | 2 (2.5%) |
| Norway | 2 (2.5%) |
| Multiple European countries | 2 (2.5%) |
| **North America** | **25 (31.3%)** |
| USA | 24 (30.0%) |
| Canada | 1 (1.3%) |
| **Asia** | **25 (31.3%)** |
| South Korea | 6 (7.5%) |
| China | 5 (6.3%) |
| Taiwan | 7 (8.8%) |
| Japan | 6 (7.5%) |
| Thailand | 1 (1.3%) |
| **Oceania** | **1 (1.3%)** |
| Australia | 1 (1.3%) |
| **South America** | **1 (1.3%)** |
| Colombia | 1 (1.3%) |
| **Africa** | **0** |
| Mean age[a] | |
| Median (IQR) | 64 (56, 71) |
| 30–49 | 7 (8.8%) |
| 50–59 | 20 (25.0%) |
| 60–69 | 29 (36.3%) |
| 70–80 | 22 (27.5%) |
| Not stated | 2 (2.5%) |
| Percent male | |
| Median (IQR) | 52% (44%, 63%) |
| $\leq$ 34% | 6 (7.5%) |
| 35–44% | 15 (18.8%) |
| 45–54% | 24 (30.0%) |
| 55–64% | 16 (20.0%) |
| $\geq$ 65% | 19 (23.8%) |
| Main morbidity /reason for inclusion | |
| CKD | 21 (26.3%) |
| Diabetes | 16 (20.0%) |
| General population | 8 (10.0%) |
| Diabetic nephropathy / kidney disease | 5 (6.3%) |
| Atrial fibrillation | 5 (6.3%) |
| Multiple CKD risk factors | 2 (2.5%) |

(*Continued*)

**Table 1.** (Continued)

| Study population characteristics | N (%) |
|---|---|
| IgA nephropathy | 2 (2.5%) |
| Infections (Hepatitis C, HIV) | 3 (3.8%) |
| Transplant recipients (liver, heart) | 3 (3.8%) |
| Autoimmune diseases (lupus, IgG4 related, vasculitis) | 3 (3.8%) |
| Gout/hyperuricemia | 2 (2.5%) |
| Other* | 10 (12.5%) |
| Data source / clinical setting | |
| Multiple care settings | 23 (28.8%) |
| Primary care | 19 (23.8%) |
| Outpatient | 17 (21.3%) |
| Diabetes clinic | 6 (7.5%) |
| Renal clinic | 3 (3.8%) |
| Diabetic-renal clinic | 1 (1.3%) |
| Not specified | 7 (8.8%) |
| Hospital | 11 (13.8%) |
| Tertiary care | 6 (7.5%) |
| Not stated | 4 (5.0%) |

[a]Other morbidities/reason for inclusion were urinary system disorders, hyperkalemia, obesity, osteoporosis, primary aldosteronism, abdominal aortic aneurysm, acute renal embolism, light chain deposition disease, lung cancer and renal cancer.

countries, including South Korea (8%), China (6%), Taiwan (9%) and Japan (8%). Research in non-English-speaking countries may be missing from review. Typically (based on median), studied populations had a mean age of 64 and were 52% male, although there was substantial variation between studies in these characteristics. Most commonly studied morbidities were CKD (26%) and diabetes (20%) although research covered a range of different populations, including (non-renal) transplant recipients and specific renal diseases. 10% studied the general population, with a further 3% studying patients with general risk factors for CKD. Clinical settings of retrieved databases varied widely, including primary care (23%), un-specified hospital settings (14%), outpatient clinics (21%), and 29% of studies used linked data across multiple care settings.

## Study methodology

Study methodology is summarised in Table 2 and a listing of key items by study is also provided in the supplementary materials S4 Table. Use of EHRs for observational research increased rapidly in recent years, with 74% of reviewed studies published in the last 5.5 years. The overwhelming majority of research was focussed on risk factor identification and causal inference (82%), with only a handful of studies attempting risk prediction (9%). Other aims included estimation of incidence or prevalence (4%) and descriptive characterisations of changes in renal function (4%). Sample size ranged drastically from 24 up to 1,597,629, with a median sample size of 1,114.

eGFR was the most commonly used measure of renal function (94%). Measures of change in renal function and methods of derivation were highly variable. Regression of absolute changes in eGFR was most common (26% of studies), although methods varied with many using mixed models but others using individual linear regression. Calculation of

**Table 2. Study methodology (N = 80).**

| Study methodology features | N (%) |
|---|---|
| Date of publication | |
| 2015–2021 | 59 (73.8%) |
| 2010–2014 | 14 (17.5%) |
| 2005–2009 | 6 (7.5%) |
| 2000–2004 | 1 (1.3%) |
| Study design | |
| Retrospective cohort study | 74 (92.5%) |
| Cross-sectional study | 4 (5.0%) |
| Case-control study | 2 (2.5%) |
| Research aims | |
| Risk factor identification / causal inference | 65 (81.3%) |
| Risk prediction | 7 (8.8%) |
| Estimation of incidence/prevalence | 3 (3.8%) |
| Descriptive characterisation of changes in renal function | 3 (3.8%) |
| Identification of sub-populations | 1 (1.3%) |
| Audit of care provision | 1 (1.3%) |
| Sample size | |
| Median (IQR) | 1114 (209, 9876) |
| ≤ 99 | 10 (12.5%) |
| 100–499 | 18 (22.5%) |
| 500–999 | 11 (13.8%) |
| 1,000–9,999 | 22 (27.5%) |
| ≥ 10,000 | 19 (23.8%) |
| Measure of renal function | |
| **eGFR** | **75 (93.8%)** |
| MDRD | 33 (41.3%) |
| CKD-EPI | 28 (35.0%) |
| MDRD, CKD-EPI combination | 1 (1.3%) |
| Taiwan CKD-EPI | 1 (1.3%) |
| Japanese formula | 3 (3.8%) |
| Not specified | 9 (11.3%) |
| **Estimated creatinine clearance** | **2 (2.5%)** |
| Cockcroft and Gault | 2 (2.5%) |
| **Serum creatinine** | **2 (2.5%)** |
| **Inverse serum creatinine** | **1 (2.5%)** |
| Measure of change in renal function over time[a] | |
| **eGFR** | **75 (93.8%)** |
| Regression slope (absolute changes) | 20 (25.0%) |
| Individual linear regression | 8 (10.0%) |
| Linear mixed model | 10 (12.5%) |
| Growth model | 1 (1.3%) |
| Generalised estimating equations | 1 (1.3%) |
| Regression slope (absolute and percent changes) | 1 (1.3%) |
| Linear mixed model | 1 (1.3%) |
| Rate of change between measures | 5 (6.3%) |
| Rate of change, not clearly defined | 4 (5.0%) |
| Rate of percentage change, not clearly defined | 3 (3.8%) |

*(Continued)*

**Table 2.** (Continued)

| Study methodology features | N (%) |
|---|---|
| Raw absolute change from baseline | 10 (12.5%) |
| Raw percent change from baseline | 13 (16.3%) |
| Raw percent change between measures | 1 (1.3%) |
| Binary progression to threshold eGFR | 6 (7.5%) |
| Binary progression (changes/threshold combination) | 3 (3.8%) |
| Transition between CKD stages | 6 (7.5%) |
| Trajectory shape class (mixed model) | 1 (1.3%) |
| Model predicted percent change per year | 1 (1.3%) |
| Model predicted eGFR at multiple time points | 1 (1.3%) |
| **Estimated creatinine clearance** | **2 (2.5%)** |
| Regression slope (absolute scale) | 1 (1.3%) |
| Raw percent change from baseline | 1 (1.3%) |
| **Serum creatinine** | **2 (2.5%)** |
| Raw absolute change from baseline | 1 (1.3%) |
| Binary progression to threshold serum creatinine | 1 (1.3%) |
| **Inverse serum creatinine** | **1 (1.3%)** |
| Regression slope (absolute changes) | 1 (1.3%) |
| Change in renal function as outcome or exposure | |
| Outcome | 74 (92.5%) |
| Exposure (if exposure, outcome listed below) | 6 (7.5%) |
| Referral to renal care | 1 (1.3%) |
| CV events | 1 (1.3%) |
| Multiple outcomes (CV, hospitalisation, death) | 1 (1.3%) |
| Advanced CKD (stage 4) | 1 (1.3%) |
| Bleeding events | 1 (1.3%) |
| Duration of follow up for renal function changes | |
| Median (IQR), years | 3.0 (1.6, 4.4) |
| < 1 year | 7 (8.8%) |
| 1–4.9 years | 48 (60.0%) |
| 5–9.9 years | 14 (17.5%) |
| ≥ 10 years | 1 (1.3%) |
| Not stated | 10 (12.5%) |
| Minimum number of renal function measures for inclusion | |
| 0 | 1 (1.3%) |
| 1 | 7 (8.8%) |
| 2 | 24 (30.0%) |
| 3 | 15 (18.8%) |
| 4 | 5 (6.3%) |
| 5 | 1 (1.3%) |
| 6 | 4 (5.0%) |
| Not stated | 23 (28.8%) |
| Percentage of target population used in analysis | |
| <50% | 17 (21.3%) |
| 50% - 75% | 9 (11.3%) |
| 75% - 90% | 5 (6.3%) |
| 90% - 95% | 5 (6.3%) |
| >95% | 8 (10.0%) |

(*Continued*)

**Table 2.** (Continued)

| Study methodology features | N (%) |
|---|---|
| Not available | 36 (45.0%) |
| Percentage of study population lost to follow up | |
| < 25% | 2 (2.5%) |
| 25% - 50% | 3 (3.8%) |
| > 50% | 1 (1.3%) |
| Not available | 62 (77.5%) |
| Complete case analysis (only including records of people with follow-up data) | 11 (13.8%) |
| Statistical tools used[b] | |
| Descriptive results only | 5 (6.3%) |
| Simple statistical tests | 9 (11.3%) |
| Linear regression models | 8 (10.0%) |
| ANOVA/ANCOVA | 2 (2.5%) |
| Kaplan-Meier estimation / life table analysis | 3 (3.8%) |
| Generalised linear models (GLMs) | 11 (13.8%) |
| Cox proportional hazards regression | 18 (22.5%) |
| Competing risks survival models | 3 (3.8%) |
| Mixed modelling methods | 12 (15.0%) |
| Other latent variable methods | 2 (2.5%) |
| Generalised estimating equations (GEEs) | 2 (2.5%) |
| Joint longitudinal survival modelling | 2 (2.5%) |
| Structural equation modelling | 1 (1.3%) |
| Multiple imputation | 5 (6.3%) |
| Machine learning methods | 3 (3.8%) |
| Statistical model used[b] | |
| **Risk factor identification / causal inference** | **N = 65** |
| Difference in means t-test | 2 (3.1%) |
| Mean difference paired t-test | 4 (6.2%) |
| Simple non-parametric tests (Mann-Whitney U) | 1 (1.5%) |
| Difference in proportions chi-squared test | 2 (3.1%) |
| ANOVA | 1 (1.5%) |
| ANCOVA | 1 (1.5%) |
| Linear regression | 7 (10.8%) |
| Logistic regression | 10 (15.4%) |
| Kaplan Meier estimation /life table analysis | 3 (4.6%) |
| Cox proportional hazards regression | 16 (24.6%) |
| Competing risk survival models | 3 (4.6%) |
| Linear mixed model | 10 (15.4%) |
| Generalised estimating equations (GEEs) | 2 (3.1%) |
| Joint longitudinal survival model | 2 (3.1%) |
| Structural equation modelling | 1 (3.1%) |
| **Risk prediction** | **N = 7** |
| Kalman filter (time series model) | 1 (14.3%) |
| Naïve Bayes classifier | 1 (14.3%) |
| Logistic regression | 4 (57.1%) |
| Cox proportional hazards regression | 1 (14.3%) |
| Random forest regression | 2 (28.6%) |
| Linear mixed model | 1 (14.9%) |

(*Continued*)

**Table 2.** (Continued)

| Study methodology features | N (%) |
|---|---|
| **Estimation of incidence/prevalence** | **N = 3** |
| Crude estimation | 3 (100%) |
| **Identification of sub-populations** | **N = 1** |
| Trajectory clustering using latent variables | 1 (100%) |
| **Audit of care provision** | **N = 1** |
| Linear mixed model | 1 (100%) |

[a]More specific details of measures of changes in renal function in individual studies assessing CKD progression and corresponding statistical analysis methods are shown in Table 4, including where time-to-event models were used in the presence of unequal follow up or censoring.

[b]Multiple items possible for a single study but focus only on main analysis of CKD progression.

absolute changes and percent changes in eGFR were also common (14% and 17% respectively), but duration of follow up varied substantially between studies. Other less common measures were rates of change calculated between measures, regression slopes on the percent scale, and binary measures for progression to thresholds of eGFR or CKD stages. 7 studies (9%) analysed rates of change in eGFR that were not clearly defined as either regression slopes or rates of change between measures. Other renal function measures studied were Cockcroft and Gault estimated creatinine clearance (3%), serum creatinine (3%) and inverse serum creatinine (1%).

Most studies (93%) analysed changes in renal function as an outcome, with only 6 studying changes in renal function as an exposure. Typical (median) duration of follow up for renal function was 3 years, but ranged from 3 months to 14 years, and was not stated in 13% of studies. Duration of follow up also commonly varied significantly between patients within individual studies, mostly due to variation in data completeness with regards to availability and timing of serum creatinine test results on the health record. Inclusion criteria relating to availability of repeat eGFR measures varied and was commonly not stated (29%). The percentage of the target population analysed could not be calculated for 36 studies (45%) due to insufficient data (Fig 2A). The study population constituted less than 50% of patients in the target population for 17 studies (21%), and less than 75% of the target population in 26 studies (33%) (Fig 2B). Statistics on data completeness were rarely stated explicitly and were often difficult to ascertain. Rates of loss to follow up were even more difficult to ascertain, and many studies sampled patients on the basis of varying levels of completeness of follow up. In 11 studies (14%), quantifying the impact of loss to follow up was not possible due to sampling based on complete follow up, and in 62 studies (78%) no data was reported on losses to follow up. The supplementary listing of individual studies provides a more detailed breakdown of analysis criteria, percentage of target population analysed and rates of loss to follow up.

Statistical methods for analysing CKD progression depended on whether the renal function measure was continuous (e.g. rate of change in eGFR) or binary (e.g. >30% change in eGFR from baseline at repeat measurement), which varied between studies. Most commonly used statistical methods were linear mixed models, linear regression, logistic regression, and Cox proportional hazards regression. Many studies used simple statistical tests, despite the inability of these methods to adjust for confounders commonly present in observational data. More sophisticated methods taking into account differential drop-outs due to death were rare. 2 studies used joint longitudinal survival models and 3 studies used competing risks survival models.

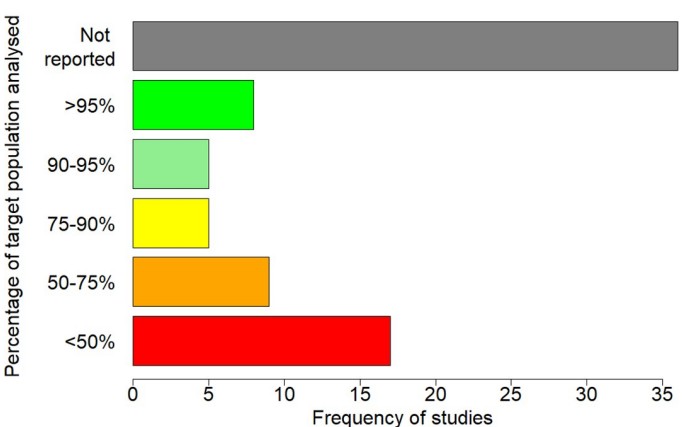 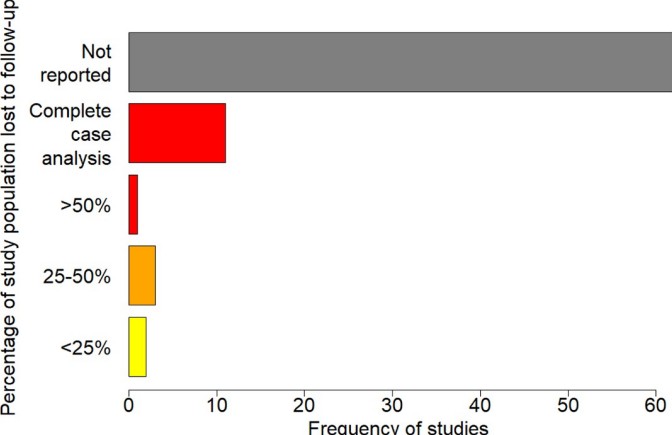

**Fig 2.** Risk of selection bias (A) and ascertainment bias (B) in individual studies.

### Handling of data quality issues and methodological challenges

Table 3 summarises how data quality issues and methodological challenges were dealt with in reviewed articles. EHR databases used for analysis rarely had good quality data on renal function, i.e. collected regularly over time and completely for all patients in the target population. A few studies attempted to improve sample completeness, for example by using imputation methods to avoid exclusions. Studies selected patients for analysis on the basis of varying levels of data completeness, relating to number of measures and duration of follow up, and many studies would have excluded patients from analysis completely on the basis of insufficient data over time. 64% of studies at least partially acknowledged this as introducing bias, 18% provided some data on sample completeness without acknowledging implications and 16% did not mention sample completeness or representativeness at all. Very few studies mentioned losses to follow up during the study period or potential reasons for loss to follow up and 61% of studies did not mention the issue of informative censoring at all. Only 6 studies (8%) tackled the issue methodologically, for example by accounting for the competing risk of death through joint longitudinal survival models and competing risks survival models.

Most studies (59%) did not mention (or tackle) the issue of missing longitudinal data on renal function tests over time. One in 6 studies did however use mixed modelling methods (16%) which may partially deal with the issue. 4 studies (5%) attempted to deal with missing longitudinal data through imputation methods. 40% of studies failed to mention missing covariate data despite covariate analysis, while 20% did not perform covariate adjustment. 25% at least partially acknowledged the issue and 16 studies (20%) made some attempt to handle missing covariate data through imputation methods, data linkage or other adjustment for missingness.

Distributional checks for renal function measures were rare, with only 5 studies (6%) mentioning distributional checks or considering alternative error distributions. Regarding the issue of variability in renal function over time and within-patient correlation, 25% did not mention (or tackle) such issues at all, 40% tackled the issue methodologically, 30% partially tackled or acknowledged the issue and a further 5% fully acknowledged such issues. 21% of studies used patient random effects to account for within-patient correlation, and 28% used outcomes which are likely to identify an important and real change.

Most studies acknowledged some aspects of population heterogeneity in analyses. At the most basic level, covariate adjusted analyses were used to account for baseline differences

**Table 3. Critique of handling of data quality and methodological challenges (N = 80).**

| Handling of data quality and methodological challenges | N (%) |
|---|---|
| Representativeness of sample to target population | |
| Not mentioned | 13 (16.3%) |
| Mentioned care pathway and inclusion criteria, but not sample completeness | 2 (2.5%) |
| Mentioned sample completeness, but not implications | 14 (17.5%) |
| Partially acknowledged implications of sample completeness | 37 (46.3%) |
| Fully acknowledged implications of sample completeness | 10 (12.5%) |
| Tackled methodologically | 4 (5.0%) |
| Methods of handling[a] | |
| None | 68 (85.0%) |
| Detailed/comprehensive database of EHRs used | 5 (6.3%) |
| Multiple imputation (to avoid exclusions) | 4 (5.0%) |
| Other imputation methods (to avoid exclusions) | 3 (3.8%) |
| Handling of informative drop-outs/censoring | |
| Not mentioned | 49 (61.3%) |
| Mentioned care pathway follow up, but not losses to follow up (inc. death) | 2 (2.5%) |
| Mentioned losses to follow up, but not implications | 7 (8.8%) |
| Partially acknowledged implications of losses to follow up | 13 (16.3%) |
| Fully Acknowledged implications of losses to follow up | 3 (3.8%) |
| Tackled methodologically | 6 (7.5%) |
| Methods of handling[a] | |
| None | 71 (88.8%) |
| Complete follow up | 1 (1.3%) |
| Joint modelling of longitudinal changes and time to drop out (including death) | 2 (2.5%) |
| Sensitivity analysis in drop-outs | 1 (1.3%) |
| Competing risks survival models | 4 (5.0%) |
| Sensitivity analysis adjusting for competing risks | 1 (1.3%) |
| Handling of missing longitudinal data | |
| Not mentioned | 47 (58.8%) |
| Mentioned care pathway follow up, but not data completeness | 4 (5.0%) |
| Mentioned data completeness, but not implications | 7 (8.8%) |
| Partially acknowledged implications of data completeness | 13 (16.3%) |
| Fully acknowledged implications of data completeness | 1 (1.3%) |
| Tackled methodologically | 8 (10.0%) |
| Methods of handling[a] | |
| None | 62 (77.5%) |
| LOCF | 1 (1.3%) |
| Imputation with mean/median | 2 (2.5%) |
| Mixed modelling | 13 (16.3%) |
| Generalised estimating equations | 1 (1.3%) |
| Multiple imputation | 1 (1.3%) |
| Handling of missing covariate data | |
| Not relevant (no covariate analysis) | 16 (20.0%) |
| Not mentioned (despite covariate analysis) | 32 (40.0%) |
| Mentioned data completeness, but not implications | 2 (2.5%) |
| Partially acknowledged implications of data completeness | 17 (21.3%) |
| Fully acknowledged implications of data completeness | 3 (3.8%) |
| Tackled methodologically | 7 (8.8%) |

(*Continued*)

**Table 3.** (Continued)

| Handling of data quality and methodological challenges | N (%) |
|---|---|
| Methods of handling[a] | |
| None | 64 (80.0%) |
| LOCF | 2 (2.5%) |
| Imputation with mean | 4 (5.0%) |
| Multiple imputation | 5 (6.3%) |
| Complete data was available for all covariates | 2 (2.5%) |
| Data linkage to improve data completeness | 1 (1.3%) |
| Adjustment for missingness | 2 (2.5%) |
| Distributional checks/issues | |
| Not mentioned | 70 (87.5%) |
| Mentioned or partially addressed | 5 (6.3%) |
| Fully Acknowledged | 0 |
| Tackled | 5 (6.3%) |
| Methods of handling[a] | |
| None | 75 (93.8%) |
| Distributional checks | 4 (5.0%) |
| Consideration of alternative error distributions | 1 (1.3%) |
| Handling of within-patient correlation / variability in kidney function over time | |
| Not mentioned | 20 (25.0%) |
| Mentioned or partially addressed | 24 (30.0%) |
| Fully Acknowledged | 4 (5.0%) |
| Tackled | 32 (40.0%) |
| Methods of handling[a] | |
| None | 35 (43.8%) |
| Random effects / latent variables | 17 (21.3%) |
| Generalised estimating equations | 2 (2.5%) |
| Modelling of stochastic process | 1 (1.3%) |
| Outcome likely to identify real change | 22 (27.5%) |
| Measures capturing AKI explicitly excluded | 1 (1.3%) |
| Paired t-test | 3 (3.8%) |
| Handling of population heterogeneity | |
| Not mentioned | 1 (1.3%) |
| Mentioned or partially addressed | 36 (45.0%) |
| Fully Acknowledged | 3 (3.8%) |
| Tackled | 40 (50.0%) |
| Method of handling[a] | |
| None | 8 (10.0%) |
| Adjustment for covariates | 21 (26.3%) |
| Interaction terms | 9 (11.3%) |
| Stratified or separate/subgroup analysis | 34 (42.5%) |
| Latent classes | 1 (1.3%) |
| Random effects | 3 (3.8%) |
| ANOVA/ANCOVA | 2 (1.5%) |
| Propensity score methods | 1 (1.3%) |
| Features in machine learning classification | 1 (1.3%) |
| Handling of confounding (risk factor / causal inference analyses only) | N = 65 |
| Not mentioned | 7 (10.8%) |

(*Continued*)

**Table 3.** (Continued)

| Handling of data quality and methodological challenges | N (%) |
|---|---|
| Mentioned or partially addressed | 17 (26.2%) |
| Fully Acknowledged | 3 (4.6%) |
| Tackled | 38 (58.5%) |
| Methods of handling[a] | |
| None | 12 (18.5%) |
| Adjustment for baseline confounders | 46 (70.8%) |
| Propensity score methods | 6 (9.2%) |

[a]Methods/approaches for handling issues are listed, regardless of whether the corresponding issues were fully tackled in analysis.

between patients (26%). Other methods included stratification or subgroup analyses to study distinct populations (43%), interaction terms allowing differing trajectories of renal function according to patient characteristics (11%) and random effects (4%). For studies performing causal analyses, 59% tackled the issue of confounding, mostly through baseline adjustment. A subset (11%) did not mention (or tackle) confounding at all, with some studies performing simple statistical tests such as t-tests and chi-squared tests despite the potential for confounding by indication.

## Definitions of CKD progression

Table 4 provides a list of CKD progression measures used in individual studies, grouped by method of derivation. A listing is provided rather than aggregate summary due to the substantial variation in the way researchers defined CKD progression across the literature. Terms used included progression, rapid progression, fast progression, rapid decline, progressive decline, progressive renal impairment, renal function deterioration and worsening renal function, while some did not provide labels, simply stating the outcome as a threshold percent change in renal function for example. There is no consistency between studies in the way these terms apply to different outcomes.

## Discussion

We performed a systematic review of peer-reviewed literature studying progression of CKD using routinely collected EHR data. Handling of data quality issues was generally poor, with unclear reporting of analysis criteria, data completeness and discussion of the implications of missing data on reliability of conclusions. For studies with sufficient data, representativeness of samples to target populations was likely to be poor with large numbers of patients excluded from analysis on the basis of poor data completeness at baseline and during follow-up thereby likely introducing selection bias. Methods capable of handling missing longitudinal data and informative losses to follow up, such as joint longitudinal survival models, were only used in a minority of studies and many studies are likely to have overstated the reliability of findings and applicability to populations of interest. Measures of change in renal function and definitions of progression varied substantially between studies, revealing a lack of consensus on clinically important and statistically robust measures in the study of CKD progression.

Unlike prospective cohort studies and clinical trials which prospectively identify patients for research and take efforts to follow up patients regularly and completely over time, retrospective analysis of routine healthcare data relies on data collected for the purposes of clinical

**Table 4. Listing of CKD progression measures in reviewed articles (52 of 80 articles).**

| Methods | Rule[a] | Term | Author [ref][b] | Year | Avg follow up | Sample size | Other methods[a] |
|---|---|---|---|---|---|---|---|
| Individual linear regression | eGFR slope decline: > **3 ml/min/1.73m²/year** | Progressors | Chase HS et al. [11] | 2014 | 6 years | 481 | Naïve Bayes classifier; logistic regression |
| | eGFR slope decline: > **median (8.1) ml/min/ 1.73m²/year** | Relatively rapid eGFR decline | Wang Y et al. [12] | 2019 | 2 years | 128 | Logistic regression |
| | eGFR slope decline: > **mean (1.5) ml/min/1.73m²/ year** | Faster decline | Abdelhafiz AH et al. [13] | 2012 | 14 years | 100 | Logistic regression |
| Linear mixed model | eGFR slope decline: > **5 ml/min/1.73m²/year** | Rapid progression | Eriksen BO et al. [14] | 2006 | 3.7 years | 3,047 | Slope interactions |
| | eGFR slope decline: > **4 ml/min/1.73m²/year** | Rapid progression | Jalal K et al. [15] | 2019 | > = 3 years | 10,927 | N/A |
| | eGFR slope decline: > **3 ml/min/1.73m²/year** | eGFR slope decline | Cabrera CS et al. [16] | 2020 | 4.3 years | 30,222 | Cox PH regression |
| | eGFR slope decline: > **0 ml/min/1.73m²/year** | Progressors (vs non-progressors) | Eriksen et al. [17] | 2010 | 4 years | 1,224 | 2-level model |
| | eGFR slope decline: > **0 ml/min/1.73m²/year** | eGFR decline | Annor FB et al. [18] | 2015 | 4 years | 575 | Structural equation modelling |
| | eGFR predicted percent rate of decline: > **5% per year** | Progression | Diggle PJ et al. [19] | 2015 | 4.5 years | 22,910 | Piecewise linear mixed model |
| Absolute change between measures | eGFR drop at any time: > **10 ml/min/1.73m²** | Progression | Butt AA et al. [20] | 2018 | 3 months | 17,624 | Difference in proportions chi-squared test |
| Percent change between measures | eGFR percent drop: >**10%**; >**20%** | Progression | Singh A et al. [21] | 2015 | 1 year | 6,435 | Logistic regression |
| | eGFR percent drop: >**15%** | Progressive renal impairment | Evans RDR et al. [22] | 2018 | 5 years | 24 | Descriptive result only |
| | eGFR percent drop: >**20%** | Transient or persistent renal function decline | Jackevicius CA et al. [23] | 2021 | Approx. 1.4 years | 49,458 | Cox PH regression |
| | eGFR percent drop: >**25%** | Progression | Lai YJ et al. [24] | 2019 | 1 year | 1,620 | Cox PH regression |
| | eGFR percent drop: >**25%** (**AND** increase in CKD stage) | Progression | Vejakama P et al. [25] | 2015 | 4.5 years | 32,106 | Competing risks survival models |
| | eGFR percent drop: >**30%** | "30% decline in eGFR" | Posch F et al. [26] | 2019 | 1.4 years | 14,432 | Cox PH regression |
| | eGFR percent drop: >**30%** | Renal function decline | Hsu TW et al. [27] | 2019 | 5 years | 5,046 | Cox PH regression |
| | eGFR percent drop: >**30%** | Rapid eGFR decline | Inaguma D et al. [28] | 2020 | 2 years | 9,911 | Logistic regression; Random forest regression |
| | eGFR percent drop: >**30%** | eGFR decline | Peng YL et al. [29] | 2020 | 1.5 years | 1,050 | Cox PH regression |
| | eGFR percent drop: >**30%** | (no label) | Yao X et al. [30] | 2017 | 11 months | 9,796 | Cox PH regression |
| | eGFR percent drop: >**30%** | "Loss of eGFR >30%" | Lamacchia O et al. [31] | 2018 | 4 years | 582 | Logistic regression |
| | eGFR percent drop: >**30%** | eGFR loss | Viazzi F et al. [32] | 2018 | 4 years | 535 | Logistic regression |
| | eGFR percent drop: >**30%** | Clinically important decline | Rej S et al. [33] | 2020 | 3.1 years | 6,226 | Cox PH regression |
| | eGFR percent drop: >**30%**; 30–50%; and 50% | Progression | Yoo H et al. [34] | 2019 | 5.7 years | 478 | Kaplan meier with log-rank test |
| | eGFR percent drop: >**40%** (or RRT initiation) | RRT40 | Tangri N et al. [35] | 2021 | 3.9 years | 32,007 | Cox PH regression |
| | eGFR percent drop: >**50%** | Renal survival endpoint | Lv L et al. [36] | 2017 | 3.1 years | 208 | Cox PH regression |
| | Serum creatinine percent increase: >**50%** | Worsening renal function | Li XM et al. [37] | 2016 | 1.8 years | 44 | Descriptive results only |
| | Estimate creatinine clearance percent drop: >**0%** | Decline in creatinine clearance | Gallant JE et al. [38] | 2005 | 1 year | 658 | Descriptive results only |

(*Continued*)

**Table 4.** (*Continued*)

| Methods | Rule[a] | Term | Author [ref][b] | Year | Avg follow up | Sample size | Other methods[a] |
|---|---|---|---|---|---|---|---|
| Rate of change between measures | eGFR drop per time elapsed (assumed): **> 2.5 ml/min/1.73m$^2$/year** | Progressive GFR decline | Herget-Rosenthal S et al. [39] | 2013 | 3 years | 803 | Logistic regression |
| | eGFR drop per time elapsed: **> 3 ml/min/1.73m$^2$/year** | Rapid progression | Morales-Alvarez MC et al. [40] | 2019 | Not stated | 594 | Descriptive comparisons |
| | eGFR drop per time elapsed: **> 5 ml/min/1.73m$^2$/year** | eGFR decline | Nderitu P et al. [41] | 2014 | 9 months | 4,145 | Logistic regression |
| | eGFR drop per time elapsed: **> 5 ml/min/1.73m$^2$/year** | Fast progression | Koraishy FM et al. [42] | 2017 | Not stated | 2,170 | Logistic regression |
| | eGFR drop per time elapsed (assumed): **> 5 ml/min/1.73m$^2$/year** | Progressive CKD | Johnson F et al. [43] | 2015 | Not stated | 200 | Difference in proportions chi-squared test |
| | eGFR drop per time elapsed: **> 5 ml/min/1.73m$^2$/year** | Rapid decline | Chakera A et al. [44] | 2015 | 7 years | 147 | Logistic regression |
| | eGFR percent drop per time elapsed (assumed): **>5% per year** | Rapid kidney function decline | Chen H et al. [45] | 2014 | 3 years | 365 | Logistic regression |
| Change in CKD stage, based on measures | Population: incident CKD stage 3 (2 x eGFR < 60 over > 3 months); Outcome: **2 x eGFR <30 over >3 months** | CKD progression from stage 3 to 4 | Perotte A et al. [46] | 2015 | Not stated | 2,908 | Cox proportional hazards regression |
| | Increase in CKD stage: **By one or more stages** | Worsening in CKD stage | Cummings DM et al. [47] | 2011 | 7.6 years | 791 | Logistic regression |
| | Increase in CKD stage: **By one or more stages (eGFR values or diagnostic codes)** | Declining kidney function | Horne L et al. [48] | 2019 | Not stated | 195,178 | Crude estimation of incidence rate |
| | Increase in CKD stage: **By one or more stages (eGFR values or coded RRT)** | CKD stage worsening | Robinson DE et al. [49] | 2021 | Approx. 3.7 years | 19,324 | Competing risks survival models |
| | Increase in CKD stage: **By one stage** | Progression of kidney dysfunction to next CKD stage | Nicolos GA et al. [50] | 2020 | 5 years | Approx 37,000 | Life-table analysis |
| | Increase in CKD stage / risk category: **To very high risk category (eGFR <30 and proteinuria (-); eGFR <45 and proteinuria (±); eGFR < 60 and proteinuria (+))** | Diabetic kidney disease progression | Yanagawa T et al. [51] | 2021 | 6.2 years | 681 | Cox PH regression |
| | Change in CKD stage: **From and to any stage, summarised by initial and final stage** | Transition between CKD stages | Vesga JI et al. [52] | 2021 | 6-month intervals | 1,783 | Crude estimation |
| Binary progression to threshold value | Threshold eGFR: **median eGFR < 30, for at least 3 consecutive months** | Nephrotoxicity | Oetjens M et al. [53] | 2014 | 8.8 years | 115 | Cox PH regression |
| | Threshold eGFR: **2 x eGFR<30 over ≥90 days with no intermediate eGFR>30** | Advanced CKD | Neuen BL et al. [54] | 2021 | 2.9 years | 91,319 | Cox PH regression |
| | Threshold eGFR: **2 x eGFR<30 over ≥90 days with no intermediate eGFR>30 (or a stage 4–5 code)** | Incident CKD stages 4–5 | Weldegiorgis M et al. [55] | 2019 | 7.5 years | 1,397,573 | Cox PH regression |
| | Threshold eGFR: **< 45 ml/min/1.73m$^2$** | Progression to CKD stage 3b | Niu SF et al. [56] | 2021 | 3.0 years | 3,114 | Cox PH regression |
| | Threshold eGFR: **< 15 ml/min/1.73m$^2$** | Renal survival endpoint | O'Riordan A et al. [57] | 2009 | 3.2 years | 54 | Kaplan meier estimation; log-rank test |
| | Threshold eGFR: **ESRD (eGFR<15 or dialysis)** | Progression to ESRD | Tsai CW et al. [58] | 2017 | 4.2 years | 739 | Cox PH regression |
| Binary progression (changes/threshold combination) | eGFR percent drop: **>50%** **AND** Threshold eGFR: **2 x eGFR <30** | Renal event | Leither MD et al. [59] | 2019 | 5.3 years | 196,209 | Cox PH regression |
| | eGFR percent drop: **>50%** **OR** Threshold eGFR: **ESRD** | "ESRD or an irreversible reduction in eGFR" | Liu D et al. [60] | 2019 | 3.7 years | 455 | Cox PH regression |
| | eGFR percent drop: **>50%** **OR** Threshold eGFR: **ESRD** | CKD progression | Rincon-Choles H et al. [61] | 2017 | 2.8 years | 1,676 | Competing risks survival models |

(*Continued*)

**Table 4.** (Continued)

| Methods | Rule[a] | Term | Author [ref][b] | Year | Avg follow up | Sample size | Other methods[a] |
|---|---|---|---|---|---|---|---|
| Latent class non-linear mixed models | Prediction of latent eGFR **trajectory class**, 6 categories | Trajectory category* | VanWagner LB et al. [62] | 2018 | 1 year | 671 | Logistic regression, conditional on class |

[a]In time-to-event analyses (e.g. Cox PH regression, competing risks survival models), the rule for progression can be met at any time during data collection, utilising repeated test results over time. In binary analyses (e.g. logistic regression), the rule is applied once per patient, likely at a specific time which may vary between studies.

[b]For consistency, article reference numbers [ref] also match those provided in the supplementary S3 File listing of reviewed studies.

care. While monitoring guidelines may be in place in healthcare systems that aim to ensure regular follow up of patients at risk of CKD progression, such guidelines may be followed at the discretion of healthcare providers, and frequency of testing and time between tests is likely to be influenced by patient risk. If patients are sampled for analysis on the basis of threshold levels of data completeness over time, there is a risk of disproportionately including patients in analysis that are followed up more regularly as a result of their evolving risk profile (selection bias) and that remain both alive and free of RRT long enough to meet the follow up criteria (survival bias). In addition, if data is collected in a single care setting but patients are managed in different care settings based on their risk, data may be informatively missing where patients move between care settings (ascertainment bias). It is highly likely that studies using EHRs that exclude patients from analysis due to poor data completeness or fail to follow up patients equally among different risk groups will have unreliable results, and results may reflect an unknown subgroup of the target population. The use of such studies to inform clinical decision-making may therefore fail to benefit the community as hoped.

There are a number of methodological challenges in longitudinal analysis of renal function that are not necessarily specific to EHRs but that are important considerations for researchers, discussed in more detail in [10, 63] and introduced earlier. In the absence of acute kidney injury, mixed effects models with patient random effects may improve estimation of changes over time compared to individual linear regressions which may lead to more extreme slope estimations. Such models allow sharing of information between patients, assuming a common mean trajectory, and they allow patients to be included in analysis with variable levels of data completeness to avoid excluding patients from analysis unnecessarily. Other benefits are the ability to perform the entire analysis (comparing exposures and outcomes) in a single model, without the loss of information and under-estimation of standard errors that may result from a 2-step model that estimates individual changes prior to further modelling. CKD is a heterogeneous disease, with various possible contributing causes and pathways of progression. Linear mixed models typically assume a common mean trajectory but other methods are available if this assumption is too strong. While random slope models allow individual trajectories to vary around a common mean slope, more sophisticated models such as latent class mixed models allow modelling of trajectory groups which may be linear or non-linear and correspond to sub-populations of patients. Another challenge is competing risk of mortality and how to handle the initiation of RRT in the analyses of repeated renal function tests, where such events are likely to be associated with rate of decline. An analysis that does not account for informative censoring may lead to biased results. Joint longitudinal survival models and competing risks survival models can be used to account for competing risks if data is available (this may require data linkage to external databases to obtain information on competing event dates).

A major finding of this review was the extreme variation in definitions of CKD progression used, and the clinical importance of each definition was unclear. More work has been done in

the last decade to identify clinically important measures of progression of CKD. In 2012, the United States Food and Drug Administration (FDA) commissioned research to identify new endpoints of CKD progression for use in clinical trials [64, 65]. Definitions were developed using data from the Chronic Kidney Disease Prognosis Consortium (CKD-PC) that showed strong association with important clinical outcomes of progression to ESRD and all-cause mortality, including thresholds of reduction in eGFR between measures of 30% and 40% over approximately 2 years, stratified by baseline eGFR. Further research that aims to define new outcomes of smaller clinically meaningful changes in renal function would be useful, as this may enable earlier identification of progression of CKD that would be useful in clinical practice, and future EHR studies could adopt such outcomes for research.

Strengths of this review include the large number of databases utilised and studies reviewed and detailed data extraction efforts, allowing a comprehensive evaluation of how well data quality issues were handled and acknowledged. The review was however limited to peer-reviewed articles and those that clarified in their abstract that repeated renal function tests were used in analysis. Limitations include the limitation to articles written in English, lack of inclusion of grey literature and issues with ascertaining whether EHRs were used as opposed to other methods of extraction from paper records. Despite this, the majority of data issues present will be the same regardless of whether electronic or paper records were used. Retrospective studies using traditional paper records will suffer from the same problems as those using electronic health records: incomplete records, variation in logging practices, addressing AKI when modeling CKD progression, loss to follow-up and competing risks.

## Conclusions

Many studies using EHRs to study progression of CKD do not fully acknowledge the biases that result from poor data quality inherent in EHRs and reporting was poor. While some studies have defined CKD progression measures similar to those validated by FDA in 2012 [64, 65] showing an understanding of identifying clinically important changes in renal function, recommendations following the systematic review by Boucquemont et al. review in 2014 [10] have not been implemented on a broader scale. Observational studies using EHRs should follow the Strengthening the Reporting of Observational Studies in Epidemiology (STROBE) [66, 67] and REporting of studies Conducted using Observational Routinely-collected health Data (RECORD) [68] guidelines, which aim to improve transparency and clarity in reporting of research. Research publications should clearly state the care pathway and intended follow up framework, data completeness eligibility criteria, the percentage of the target population excluded based on those criteria, whether there were differences in characteristics of those included vs. excluded and according to important risk factors, as well as rates of loss to follow up. Where possible, researchers should attempt to ascertain reasons for loss to follow up, which may involve linkage to external data. Researchers should consider using existing validated outcomes of CKD progression and we hope that heterogeneity in definitions of CKD progression will improve over time. Focussing research questions on populations for which regular data collection is performed as part of routine care may offer a route to better quality data on changes of renal function over time and important changes in renal function will be easier to identify accurately in patients with reduced renal function at baseline, such as those with established CKD where GFR-estimating equations perform better.

## Supporting information

**S1 File. PRISMA checklist.**
(DOC)

**S2 File. MEDLINE database search strategy.**
(DOCX)

**S3 File. List of reviewed studies.**
(DOCX)

**S1 Table. Summary of study populations, where unclear if EHRs used.**
(DOCX)

**S2 Table. Study methodology, where unclear if EHRs used.**
(DOCX)

**S3 Table. Critique of handling of data quality and methodological challenges, where unclear if EHRs used.**
(DOCX)

**S4 Table. Listing of key features of all included studies, sorted by year of publication.**
(DOCX)

**S1 Data. Data extraction spreadsheet.**
(XLSX)

## Acknowledgments

Only the listed authors contributed to the work reported in this manuscript.

## Author Contributions

**Conceptualization:** Faye Cleary, David Prieto-Merino, Dorothea Nitsch.

**Data curation:** Faye Cleary.

**Funding acquisition:** Faye Cleary.

**Investigation:** Faye Cleary.

**Methodology:** Faye Cleary, David Prieto-Merino, Dorothea Nitsch.

**Project administration:** Faye Cleary.

**Supervision:** David Prieto-Merino, Dorothea Nitsch.

**Validation:** David Prieto-Merino, Dorothea Nitsch.

**Writing – original draft:** Faye Cleary.

**Writing – review & editing:** David Prieto-Merino, Dorothea Nitsch.

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
