## [Decision Letter · Decision Letter 0]

11 May 2021

PONE-D-21-03779

A systematic review of statistical methodology used to evaluate progression of chronic kidney disease using electronic healthcare records

PLOS ONE

Dear Dr. Cleary,

Thank you for submitting your manuscript to PLOS ONE. After careful consideration, we feel that it has merit but does not fully meet PLOS ONE’s publication criteria as it currently stands. Therefore, we invite you to submit a revised version of the manuscript that addresses the points raised during the review process.

The authord need to address reviewers' comments.

Reviewer # 1:

The systematic analysis by Faye Cleary et al. is useful in terms of identifying the opportunities coming from increasing availability of electronic health records, as well as pointing to common mistakes and challenges associated with their intrinsic nature. In this regard, the authors did a good job summarizing the methodology across the 65 included studies. It would be nice to note that retrospective studies using traditional paper records will suffer from the same problems as those using electronic health records: incomplete records, variation in logging practices, addressing AKI when modeling CKD progression, loss to follow-up and competing risks. I don't have other concerns about this work.

Reviewer # 2:

A well written systematic review with proper design, presentation of result, and discussion. I have few comments:

-The search needs to be updated beyond 7th May 2020. Multiple studies were published after this date.

-Clarify the study methodology: Sample size (before and after exclusions for reasons of data completeness);

-The authors stated that “It is likely that if research is missing from review, then data quality issues in missing studies are likely to be of a similar quality or worse quality than studies reviewed”.  Only studies reported in English language were included in this review. Therefore,  it is unfair to label studies in non-English language as inferior in quality.

-What is the outcome for Cox proportional hazards regression? I assume time to event !!

We look forward to receiving your revised manuscript.

Kind regards,

Stanislaw Stepkowski

Academic Editor

PLOS ONE

Additional Editor Comments:

The authord need to address reviewers' comments.

Reviewer # 1:

The systematic analysis by Faye Cleary et al. is useful in terms of identifying the opportunities coming from increasing availability of electronic health records, as well as pointing to common mistakes and challenges associated with their intrinsic nature. In this regard, the authors did a good job summarizing the methodology across the 65 included studies. It would be nice to note that retrospective studies using traditional paper records will suffer from the same problems as those using electronic health records: incomplete records, variation in logging practices, addressing AKI when modeling CKD progression, loss to follow-up and competing risks. I don't have other concerns about this work.

Reviewer # 2:

A well written systematic review with proper design, presentation of result, and discussion. I have few comments:

-The search needs to be updated beyond 7th May 2020. Multiple studies were published after this date.

-Clarify the study methodology: Sample size (before and after exclusions for reasons of data completeness);

-The authors stated that “It is likely that if research is missing from review, then data quality issues in missing studies are likely to be of a similar quality or worse quality than studies reviewed”. Only studies reported in English language were included in this review. Therefore, it is unfair to label studies in non-English language as inferior in quality.

-What is the outcome for Cox proportional hazards regression? I assume time to event !!

Journal Requirements:

3. Please remove your figures from within your manuscript file, leaving only the individual TIFF/EPS image files, uploaded separately.  These will be automatically included in the reviewers’ PDF.

Reviewers' comments:

Reviewer's Responses to Questions

**Comments to the Author**

1. Is the manuscript technically sound, and do the data support the conclusions?

Reviewer #1: Yes

Reviewer #2: Yes

2. Has the statistical analysis been performed appropriately and rigorously? 

Reviewer #1: Yes

Reviewer #2: Yes

3. Have the authors made all data underlying the findings in their manuscript fully available?

Reviewer #1: Yes

Reviewer #2: Yes

4. Is the manuscript presented in an intelligible fashion and written in standard English?

Reviewer #1: Yes

Reviewer #2: Yes

5. Review Comments to the Author

Reviewer #1: The systematic analysis by Faye Cleary et al. is useful in terms of identifying the opportunities coming from increasing availability of electronic health records, as well as pointing to common mistakes and challenges associated with their intrinsic nature. In this regard, the authors did a good job summarizing the methodology across the 65 included studies. It would be nice to note that retrospective studies using traditional paper records will suffer from the same problems as those using electronic health records: incomplete records, variation in logging practices, addressing AKI when modeling CKD progression, loss to follow-up and competing risks. I don't have other concerns about this work.

Reviewer #2: A well written systematic review with proper design, presentation of result, and discussion. I have few comments:

-The search needs to be updated beyond 7th May 2020. Multiple studies were published after this date.

-Clarify the study methodology: Sample size (before and after exclusions for reasons of data completeness);

-The authors stated that “It is likely that if research is missing from review, then data quality issues in missing studies are likely to be of a similar quality or worse quality than studies reviewed”. Only studies reported in English language were included in this review. Therefore, it is unfair to label studies in non-English language as inferior in quality.

-What is the outcome for Cox proportional hazards regression? I assume time to event !!

6. PLOS authors have the option to publish the peer review history of their article (what does this mean?). If published, this will include your full peer review and any attached files.

Reviewer #1: **Yes: **Dulat Bekbolsynov

Reviewer #2: **Yes: **Sadik A. Khuder

---

## [Author Response · Author response to Decision Letter 0]

26 Oct 2021

I respond to specific reviewer comments individually below. I break this down point by point for reviewer 2 using the word 'RESPONSE' for each individual point separately. 

Comments reviewer #1:

The systematic analysis by Faye Cleary et al. is useful in terms of identifying the opportunities coming from increasing availability of electronic health records, as well as pointing to common mistakes and challenges associated with their intrinsic nature. In this regard, the authors did a good job summarizing the methodology across the 65 included studies. It would be nice to note that retrospective studies using traditional paper records will suffer from the same problems as those using electronic health records: incomplete records, variation in logging practices, addressing AKI when modeling CKD progression, loss to follow-up and competing risks. I don't have other concerns about this work.

RESPONSE:

We are pleased that the reviewer sees the value and quality of our work. In response to reviewer suggestions, we have updated text in the discussion to note that traditional paper records will suffer from the same problems as those using electronic healthcare records. 

Comments Reviewer #2:

A well written systematic review with proper design, presentation of result, and discussion. 

RESPONSE: We are pleased that the reviewer believes we have conducted a well-designed and presented review of the literature.

-The search needs to be updated beyond 7th May 2020. Multiple studies were published after this date. 

RESPONSE: We have updated the search dates to include studies available in the 4 databases covered by the review as of August 2021, allowing us to capture more recently published studies.

-Clarify the study methodology: Sample size (before and after exclusions for reasons of data completeness);

RESPONSE: We have clarified in the methods text that data completeness inclusion criteria refer to the specific study inclusion criteria applied prior to main analyses being performed that aimed to restrict analyses to only those patients with sufficient data completeness to be deemed suitable for analysis, with such criteria expected to vary between studies. The explanation of the calculation for “percent of target population analysed” also shows readers how we used sample size data before and after data completeness inclusion criteria were applied to uncover the extent to which patients were excluded from analysis purely due to failure to meet a study’s data completeness requirements.

-The authors stated that “It is likely that if research is missing from review, then data quality issues in missing studies are likely to be of a similar quality or worse quality than studies reviewed”. Only studies reported in English language were included in this review. Therefore, it is unfair to label studies in non-English language as inferior in quality.

RESPONSE: Our comment that “With peer-reviewed literature expected to go through certain research quality checks, it is likely that if research is missing from review, then data quality issues in missing studies are likely to be of a similar quality or worse quality than studies reviewed” was intended to convey that studies missing from review due to not being peer-reviewed are likely to be of similar or worse quality as/than those peer-reviewed, due to the quality checks that peer-reviewed studies go through. It was not intended to say anything about studies published in non-English languages. We have clarified in the methods text that we anticipate that studies published in both English and non-English languages are likely to be of similar quality.

-What is the outcome for Cox proportional hazards regression? I assume time to event !!

RESPONSE: I’m not 100% sure where exactly in the manuscript the reviewer is referring to in this comment, but I imagine it may be results Tables 2 and 4. I would like to clarify what is reported and what is not. Due to the anticipated variation in how researchers define progression of kidney disease over time and the challenges this may pose in clinical interpretability of findings of research studies, a key aim of our review was to summarise how researchers measured changes in renal function over time. We also reported methods for analysis (which include as the reviewer states Cox proportional hazards regression models with such models using as outcome time to some event). In our reporting of study methodology (Table 2), we summarise “Measure of change in renal function over time”. As an example, an event of a 30% decline in eGFR between measures would be reported as “Raw percent change in eGFR between measures” (as this captures how changes over time were measured) and we do not specifically state whether this was analysed as time to event or as a binary outcome but we do report the method of analysis (“Statistical model used”), for example Cox proportional hazards regression. Table 4 further clarifies precise measures of changes in renal function over time for each individual study (e.g. percent loss in eGFR between measures >30%) alongside the methods used (e.g. Cox PH regression). Although we do not specifically state what the outcome is (e.g. time to percent loss in eGFR between measures >30%), this is inferred. We have added a comment to Table 4 to clarify this.

---

## [Decision Letter · Decision Letter 1]

7 Feb 2022

A systematic review of statistical methodology used to evaluate progression of chronic kidney disease using electronic healthcare records

PONE-D-21-03779R1

Dear Dr. Cleary,

We’re pleased to inform you that your manuscript has been judged scientifically suitable for publication and will be formally accepted for publication once it meets all outstanding technical requirements.

Kind regards,

Mabel Aoun, MD, MPH

Academic Editor

PLOS ONE

Additional Editor Comments (optional):

Reviewers' comments:

Reviewer's Responses to Questions

**Comments to the Author**

1. If the authors have adequately addressed your comments raised in a previous round of review and you feel that this manuscript is now acceptable for publication, you may indicate that here to bypass the “Comments to the Author” section, enter your conflict of interest statement in the “Confidential to Editor” section, and submit your "Accept" recommendation.

Reviewer #1: All comments have been addressed

2. Is the manuscript technically sound, and do the data support the conclusions?

Reviewer #1: Yes

3. Has the statistical analysis been performed appropriately and rigorously? 

Reviewer #1: Yes

4. Have the authors made all data underlying the findings in their manuscript fully available?

Reviewer #1: Yes

5. Is the manuscript presented in an intelligible fashion and written in standard English?

Reviewer #1: Yes

6. Review Comments to the Author

Reviewer #1: (No Response)

7. PLOS authors have the option to publish the peer review history of their article (what does this mean?). If published, this will include your full peer review and any attached files.

Reviewer #1: **Yes: **Dulat Bekbolsynov

---

## [Editor Report · Acceptance letter]

9 Feb 2022

PONE-D-21-03779R1 

A systematic review of statistical methodology used to evaluate progression of
chronic kidney disease using electronic healthcare records 

Dear Dr. Cleary:

I'm pleased to inform you that your manuscript has been deemed suitable for publication in PLOS ONE. Congratulations! Your manuscript is now with our production department. 

Kind regards, 

on behalf of

Dr. Mabel Aoun 

Academic Editor

PLOS ONE